# Prediction of adherence to treatment with statins and anti-platelet drugs in first-year post-stroke patients: Validation of beta-regression models

Elias Edward Tannous[1,2]*, Shlomo Vinker[3], David Stepensky[1], Eyal Schwarzberg[1]

1 Department of Clinical Biochemistry and Pharmacology, Faculty of Health Sciences, Ben-Gurion University of the Negev, Beer-Sheva, Israel, 2 Pharmacy Services, Hillel Yaffe Medical Centre, Derech Hashalom, Hadera, Israel, 3 Leumit Health Care Services, Tel Aviv, Israel

* tanus@bgu.ac.il

## Abstract

Stroke is the third most common cause of disability and the second most common cause of death worldwide. Greater levels of medication adherence after stroke or transient ischemic attack are associated with improved survival. Very few medication adherence prediction models are available and have not been validated using external data. The current study aimed to evaluate the predictive performance of previously published beta regression models for statin and antiplatelet adherence at 1 year in patients' post-stroke or transient ischemic attack. The models use the first 90-day adherence data as a single predictor for 1-year adherence. Adherence was measured using the Proportion of Days Covered (PDC), which utilized prescription-filling data. Model performance was assessed using the following metrics: $R^2$ (proportion of variance explained), the difference between the mean observed and the mean predicted PDC, and the calibration slope, which ideally should be one. 2369 were included in the statin cohort, and 2147 patients were included in the antiplatelet cohort. $R^2$ was 0.67 and 0.56 for statin and antiplatelet models, respectively. The difference between the mean observed and the mean predicted PDC was −3.7% and −2.5% for statin and antiplatelet models, respectively. The calibration slopes were 1.06 and 0.96 for the statin and antiplatelet models, respectively. The model performed well on a new patient population comprised of post-stroke patients and may be used for early identification of patients at high risk for low 1-year adherence within 90 days post-stroke, enabling timely, targeted adherence-support interventions.

## Introduction

Stroke is the third most common cause of disability and the second most common cause of death worldwide [1]. A population-based study from Australia and New Zealand found that first ischemic stroke resulted in five-year loss of life expectancy when compared with matched patients from the general population [2]. Pooled

**Data availability statement:** The data underlying this article were provided by LHCS by permission. To ensure patient confidentiality the data availability was restricted by the ethical committee of LHCS. De-identified datasets containing variables necessary to reproduce the analysis can be shared upon request. Researchers seeking access to the data should contact LHCS ethics committee coordinator Mrs. Alice Pincu [apincu@leumit.co.il].

**Funding:** The author(s) received no specific funding for this work.

**Competing interests:** The authors have declared that no competing interests exist.

cumulative risk of stroke recurrence was 3.1% at 30 days, 11.1% at 1 year. The annual risk of stroke recurrence is higher in the first year compared to 5 or 10 years post-stroke [3].

The effectiveness of aspirin/clopidogrel and statins for secondary prevention of ischemic stroke and cardiovascular events is supported by high-quality data [4,5]. However, adherence remains a challenge. In a meta-analysis including 375,408 stroke patients from 63 observational studies, the proportions of medication adherence and persistence were 64.1% and 72.2%, respectively [6]. Greater levels of medication adherence after stroke or transient ischemic attack (TIA) are associated with improved survival, even among patients with near-perfect adherence [7].

Predicting adherence to antiplatelet and statin therapy is of paramount importance, as it may help healthcare providers select patients who will benefit most from adherence-enhancing interventions, such as individualized cardiovascular education, emotional engagement with cardiovascular prevention, mobile health prompts, and improving patient acceptability of lipid-lowering therapies [8]. However, very few studies have developed prediction models for secondary prevention medicine [9–11]. The main limitation of the available models is that they dichotomize PDC at the 0.8 level, thus producing a binary adherence status [9–11]. A recent study on the determinants of adherence in this patient population found that depression and higher educational level were associated with suboptimal adherence. However, the authors did not develop a prediction model for clinical use [12]. While some published models predict adherence in post-stroke patients, none have been externally validated [9–11].

External validation of a prediction model involves evaluating its performance in a new dataset. In external validation, a previously developed model is applied to a new dataset, and the model`s predictions are tested for their accuracy. External validation provides information concerning a model's reproducibility and generalizability [13].

In a previous study, we demonstrated that Bayesian beta-regression modeling enabled the modeling of Proportion of Days Covered (PDC) as a proportion in post-Myocardial Infarction (MI) patients [14]. The models exhibited good performance, as assessed by visual posterior predictive checks and the Bayesian coefficient of determination ($R^2$). Moreover, our results suggest that a simplified model, with data on medication use in the first 90 days post-event as a single predictor of 1-year PDC, performs as well as a full model with demographic, clinical, and socioeconomic predictors [14]. Notably, more than 90% of the patients in this study had 30-day refill patterns (30-day prescription cycles) [14].

The former finding may be explained by habit development [15,16]. Several studies found that habit formation is a strong predictor of long-term medication adherence [16,17]. While medication-purchasing data is not a perfect measure for habit formation, it is a convenient and easily obtained surrogate for habit formation, especially when it is not feasible to fill out questionnaires that measure habit strength [18]. Therefore, although originally developed on post-MI patients, a predictive model using the first 90 days post-event (a surrogate for habit formation) as a single predictor of 1-year PDC may be expected to perform well on a new population with a different disease than the one on which it was originally developed.

In the current study, we performed an external validation of a previously published model on a new patient population comprised of post-ischemic stroke patients.

## Methods

Minimum sample size for external validation was calculated according to the following four criteria [19]. The criterion with highest needed sample size was defined as the minimal sample size needed for external validation.

Criterion (i): Calculate the sample size needed to precisely estimate variance explained in the external validation data-set ($R^2_{val}$)

$$n = \frac{4R^2_{val}(1 - R^2_{val})^2}{SE^2_{\hat{R}^2_{val}}}$$

Based on a previously published study [14], we assumed an $R^2$ value of 0.5. We assumed a standard error (SE) = 0.0255 based on the recommendations by Archer et al. [19]. According to the above formula, the minimum sample size needed is $768.93 \approx 769$ patients.

Criterion (ii): Calculate the sample size required to precisely estimate calibration-in-the-large (i.e., the difference between the predicted and observed mean PDC)

$$n = \frac{var\left(Y_i\right)\left(1 - R^2_{val}\right)}{SE^2_{C\hat{I}TL}}$$

Where $SE_{C\hat{I}TL}$ is the standard error for the difference between the mean predicted and mean observed PDC. We assumed $SE^2_{C\hat{I}TL} = 0.0255$ based on the recommendations by Archer et al. [19], and $var\left(Y_i\right) = 0.04$, and $R^2_{val} = 0.5$ based on the data from a previously published study [14]. According to the above formula, the minimum sample size needed is $30.75 \approx 31$ patients.

Criterion (iii): Calculate the sample size needed to precisely estimate the calibration slope.

$$n = \frac{\lambda^2_{cal}(1 - R^2_{cal})}{SE^2_{\hat{\lambda}_{cal}} R^2_{cal}} + 1$$

Where $\lambda_{cal}$ is calibration slope and $SE_{\hat{\lambda}_{cal}}$ is the standard error of calibration slope. We assumed $\lambda_{cal} = 1$ and $SE_{\hat{\lambda}_{cal}} = 0.051$, based on the recommendations by Archer et al. [19] and $R^2_{cal} = R^2_{val} = 0.5$ based on the data from a previously published study [14]. Thus, the minimum sample size needed is $393.15 \approx 394$ patients.

Criterion (iv): Calculate the sample size for precise estimation of residual variances. To ensure a 10% margin of error in residual variance estimates from the calibration models, 235 patients would be needed.

Thus, based on the criterion with the highest needed sample size, at least 769 patients are required for external validation.

## Data collection

We collected social and demographic information from the electronic medical records. The electronic medical records of Leumit Health Care Services (LHCS), one of Israel's providers of public and semi-private health services, include data from multiple sources, including records of primary care physicians, community specialty clinics, hospitalizations, laboratories, and pharmacies. LHCS is the smallest Israeli health provider, offering health services and insurance to approximately

7.6% of the Israeli population. However, The LHCS membership broadly reflects the Israeli population and comprises individuals from various ethnic, geographic, and socioeconomic backgrounds. Data on prescription filling included the date of filling, quantity supplied, and dosing information. Diagnoses were captured in the registry through diagnosis-specific algorithms, utilizing the International Classification of Diseases, Ninth Revision (ICD-9) code reading and laboratory test results. The data was accessed between June 1, 2024, and July 4, 2024. The authors did not have access to information that could identify individual participants during or after data collection. The data was coded to ensure participant anonymity. The study protocol was approved by the LHCS Ethics Committee (Protocol number: 0021–22-LEU).

## Patient cohort

Candidate patients were screened from the LHCS database using ischemic stroke and transient ischaemic attack (TIA) related ICD-9 codes (433.x1,434,V12.54) between January 2016 and May 2021. While Stroke and TIA are distinct clinical presentations, they do share a common vascular pathophysiology and are managed with identical secondary prevention strategies. Therefore, combining stroke and TIA patients increases clinical applicability and reflects real-world practice. The index date was defined as the date the event of TIA or stroke was recorded. Patients under the age of 18 and pregnant women were excluded from the study. Moreover, patients who died within 100 days post-index date and those who switched health insurance providers during the first year post-index date were excluded. For patients who had more than one event during the study period, we included only the first event.

To identify patients who may have discontinued antiplatelet therapy due to bleeding, we utilized bleeding-related ICD-9 codes to detect any bleeding events during the follow-up period [20].

## PDC calculation

PDC calculation was conducted according to the recommendations of a recently published scoping review and the TEN-SPIDERS tool [21].

The denominator was calculated as the follow-up time since the index date with a maximum value of 365.

The numerator was calculated using the following method: First, adjustment of dates and identification of gaps post-index date was performed using 'adheRenceRX' package in R, so that carry-over was granted for early refills of the same drug [22]. Subsequently, the length of total covered days was calculated from the first fill date to the last one (including the days covered of the last fill). In addition, pre-supply was accounted for using 90-day look-back and carried over into the observation period. Switching between different statins and different antiplatelets was allowed for, and the final PDC was calculated for statins as a group and antiplatelets as a group (i.e., carry-over was granted for therapeutic switches in the same medication group, e.g., switching from rosuvastatin to atorvastatin, or from aspirin to clopidogrel). Hospitalization days during the follow-up period were added to the numerator, and in-hospital supply was assumed since patients in hospitals in Israel do not use their own medications as hospitals are required to supply them, unless contraindicated. The final formula used to calculate PDC was:

$$PDC = \frac{Presupply\ days + Gap\ adjusted\ post\ index\ days + Inhospital\ days}{Minimum\ [days\ from\ index\ date\ to\ end\ of\ observation, 365]}$$

For the full TEN-SPIDERS tool, please see S1 Table (supplementary material).

## The models

In a previously published study, we developed and compared predictive models for medication adherence in post-Myocardial Infarction patients (MI) [14]. Specifically, we developed models to predict the first-year PDC in patients post-MI for three medication groups: Statins, P2Y12 inhibitors, and Aspirin. The main novelty introduced was the

modelling of PDC as a proportion using beta regression. Previous studies have dichotomized PDC (usually at the threshold of 80%) and then proceeded to predict PDC using logistic regression models. This approach oversimplifies adherence status and loses valuable information. The models developed explained approximately 50–60% of the variability [14]. Moreover, we showed that a simple model, including only data on adherence in the first 90 days post MI (first 90-days covered) as a single predictor of 1-year PDC, performed similarly to a full model including the first 90-days covered in addition to demographic, clinical and socioeconomic predictors (age, distance from primary clinic, number of daily medications, socioeconomic score(based on The Israeli Central Bureau of Statistics characterization and classification of geographical Units by the Socio-Economic Level of the Population.), country of origin (Israel/other), smoking status, depression diagnosis, use of any serotonin specific reuptake inhibitor, number of Flu vaccines administered in the last 5 years before index date, number of Leumit Healthcare services clinics per 10000 clients, number of pharmacies per 10000 clients) [14]. Importantly, the models were developed using a population where over 90% of patients refilled their prescriptions every 30 days (one month). Therefore, the structure of the models is inherently tied to 1-month prescription refills.

**Statin model.** In the previously developed statin model [14], the linear predictor for the logit-transformed 1-year PDC had an intercept of −1.65 and a slope of 0.04. The model formula is as follows:

$$logit(1year\ PDC) = -1.65 + 0.04 * number\ of\ days\ covered\ at\ 90\ days$$

To get the 1-year PDC on the original scale, the following back transformation was applied

$$1year\ PDC = \frac{1}{1 + e^{-(-1.65+0.04*number\ of\ days\ covered\ at\ 90\ days)}}$$

**Antiplatelet model.** In the previously developed aspirin model [14], the linear predictor for the logit-transformed 1-year PDC had an intercept of −1.52 and a slope of 0.04. The model formula is as follows:

$$logit(1year\ PDC) = -1.52 + 0.04 * number\ of\ days\ covered\ at\ 90\ days$$

To get the 1-year PDC on the original scale, the following back transformation was applied

$$1year\ PDC = \frac{1}{1 + e^{-(-1.52+0.04*number\ of\ days\ covered\ at\ 90\ days)}}$$

### External validation

The primary performance metrics for the model, as proposed by Archer et al. [19], were:

1- $R^2$ (the proportion of variance explained)

2- calibration-in-the-large (CITL, agreement between predicted and observed outcome values on average)

3- calibration slope (agreement between predicted and observed values across the range of predicted values)

Additionally, the Root Mean Squared Error (RMSE) and Mean Absolute Error (MAE) were calculated.

Calibration measures the agreement between predicted PDC (according to the model) and observed PDC (in the external validation dataset). The calibration model has the following form:

$$PDCobserved(i) = \propto cal + \lambda cal * (PDCpredicted\ (i)) + e(i)$$

Where $\propto cal$ is the intercept of the model, $\lambda cal$ the slope and $e(i)$ the difference between the predicted and observed PDC. A perfect model would have $\propto cal$ = 0, and $\lambda cal$ = 1. CITL can be estimated by the following equation:

$$CITL = Mean\ (observed\ PDC) -\ Mean\ (predicted\ PDC)$$

In a model where $\lambda cal$ = 1, the intercept $\propto cal$ is equal to CITL. The closer $\lambda cal$ is to one, the better the model performance. Likewise, the closer $\propto cal$ to zero, the better the model performance.

Moreover, model performance was evaluated using a calibration curve and histograms of errors and absolute errors (i.e., the difference and absolute difference between the observed and predicted 1-year PDC, respectively) for each medication group. External validation was performed in R [23].

## Results

2426 patients with ischemic stroke and transient ischemic attack (TIA) related ICD-9 codes were identified. 9 were excluded for being pregnant during the follow-up period. 74 died during the follow-up period, 23 of whom died less than 100 days after the index date and were excluded. 25 were excluded for erroneous data. From the antiplatelets cohort (patients receiving aspirin or clopidogrel), 222 patients were excluded: 221 due to additional oral anticoagulant use and one due to an intracerebral hemorrhage. Approximately 94% of patients had a 1-month supply (they filled their prescription for one month at each pharmacy visit), and 6% had a 2 or 3-month supply (they filled their prescription for two or three months at each pharmacy visit). No bleeding events were identified utilizing bleeding-related ICD-9 codes, which may indicate limited sensitivity rather than a true absence of events. Minor bleeding incidents may have occurred, but were possibly not severe enough to prompt patients to seek medical consultation.

The final cohort for statin model external validation included 2369 patients (which is substantially higher than the minimal needed sample size). The median (IQR) age was 70 (61, 79), and 51% (1208) of the participants were male. 22% (522) used antidepressants, and 5.6% (132) had diagnosed depression. The median (IQR) number of daily medications was 4 (1,8) and the median (IQR) socioeconomic score was 9 (6,12), therefore most patients included could be considered as middle-class. The median (IQR) number of days covered at 90 days was 30 (0–86), and the median (IQR) PDC at 1 year was 0.41 (0, 0.83). 64.7% were new users, and 35.3% were prevalent users. The characteristics of the statin cohort and missing values are presented in Table 1.

The final cohort for the external validation of the antiplatelets model included 2147 patients. The median (IQR) age was 70 (60, 78), and 51% (1208) of the participants were male. 21% (459) used antidepressants, and 5.6% (120) had diagnosed depression. The median (IQR) number of daily medications was 4 (1,7) and the median (IQR) socioeconomic score was 9 (6,12). The median (IQR) number of days covered at 90 days was 30 (0,90) and the median (IQR) PDC at 1 year was 0.59 (0.08, 0.86). 84.3% were new users, and 15.7% were prevalent users. The antiplatelets cohort characteristics and missing values are presented in Table 1.

### External validation

**Statin model.** For the statin model, R² was 0.67, indicating that 67% of the variance in the external validation dataset was explained by the model. Although a high R² is expected when early adherence predicts 1-year PDC, a high R² in the external validation dataset indicates that the relationship between early adherence and 1-year PDC remains consistent across different patient groups and that the model can be applied to other populations. The CITL value was −0.037, that is, a mean difference of −3.7% between observed and predicted 1-year PDC (the mean observed and predicted 1-year PDC were 42.8% and 45.9%, respectively).

The calibration slope was 1.06 (95% CI 1.03,1.09), a value that is very close to 1 and indicates good agreement between predicted and observed outcomes across the entire range of predicted values. The calibration curve is

**Table 1. Patient demographic, clinical and socioeconomic characteristics.**

| Characteristic | statin cohort | antiplatelet cohort (aspirin/clopidogrel) |
|---|---|---|
| Characteristic | N = 2,369 | N = 2,147 |
| Age (years), median (IQR) | 70 (61, 79) | 70 (60, 78) |
| Sex (Female), n (%) | 1,161 (49) | 1,049 (49) |
| 1- year PDC, median (IQR) | 0.41 (0.00, 0.83) | 0.59 (0.08, 0.86) |
| Number of days covered at 90-days post index date, median (IQR) | 30 (0, 86) | 30 (0, 90) |
| New users, n (%) | 1533 (64.7) | 1810 (84.3) |
| Depression, n (%) | 132 (5.6) | 120 (5.6) |
| Use of any antidepressant, n(%) | 522 (22) | 459 (21) |
| Number of daily medications, median (IQR) | 4 (1, 8) | 4 (1, 7) |
| Ex- smoker, n (%) | 58 (2.4) | 48 (2.2) |
| Non smoker, n (%) | 1,781 (75) | 1,597 (74) |
| Smoker, n (%) | 511 (22) | 484 (23) |
| Unknown smoking status, n | 19 (0.8) | 18 (0.8) |
| Socioeconomic score, median (IQR) | 9 (6, 12) | 9 (6, 12) |
| Unknown, n | 150 | 140 |
| Distance from primary clinic (Km), median (IQR) | 1.432 (0.875, 2.232) | 1.437 (0.866, 2.227) |
| Unknown, n | 733 | 668 |
| 1-month supply, n (%) | 2231(94.2) | 2011 (93.7) |
| 2-month supply, n (%) | 74 (3.1) | 56 (2.6) |
| 3-month supply, n (%) | 64 (2.7) | 80 (3.7) |
| PDC; Proportion of days covered | | |

presented in Fig 1. 70% of predictions had an absolute error of less than 16%. Histograms of errors and absolute errors are presented in Fig 2. RMSE was 0.229 and MAE was 0.184. We compared the model's MAE to that of a baseline model that predicts the mean outcome for all observations. The model reduced the MAE by 51.3% compared to this baseline, indicating a significant improvement in prediction accuracy. This "relative MAE reduction" is similar to the concept of mean absolute scaled error [24], and provides a clear measure of predictive usefulness. On average, the model cuts the prediction error in half compared to simply predicting the mean. Table 2 summarizes the external validation results.

**Antiplatelets model.** For the Antiplatelets model, the $R^2$ was 0.56, indicating that 55.9% of the variance in the external validation dataset was explained by the model. While a high $R^2$ is anticipated when early adherence serves as the predictor of 1-year PDC, a high $R^2$ within the external validation dataset indicates that the average relationship between early adherence and 1-year PDC remains relatively consistent across diverse patient populations, thereby implying the model's transportability. The CITL value was −0.025, representing a mean difference of −2.5% between the observed and predicted 1-year PDC (the mean observed and predicted 1-year PDC were 50.6% and 53.1%, respectively). The calibration slope was 0.96 (95% CI: 0.93,1.00), a value that is very close to 1, indicating strong agreement between predicted and observed outcomes across the entire range of predicted values. The calibration curve is presented in Fig 3. 70% of predictions had an absolute error of less than 20.9%. Histograms of errors and absolute errors are presented in Fig 4. RMSE was 0.255 and MAE was 0.202. The model reduced the MAE by 42.1% relative to a baseline model, indicating substantial predictive improvement. On average, the model reduces the prediction error by 42.1% compared to simply predicting the mean proportion. Table 2 summarizes the external validation results.

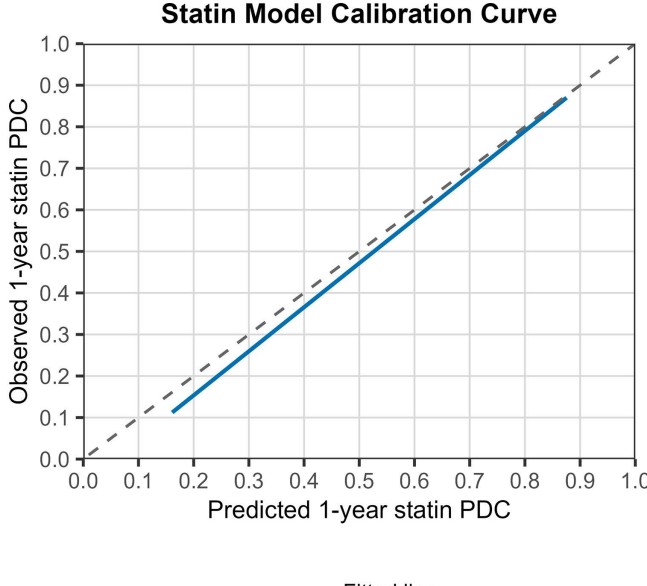

### Statin Model Calibration Curve

— Fitted line

**Fig 1. Statin model calibration curve.** Calibration performance on external validation of the prediction model for statin 1-year PDC. The 45° dashed-line shows perfect calibration and the blue line shows the predicted average PDC statin 1-year. The plot shows the observed outcome (y-axis) versus the predicted outcomes (x-axis).

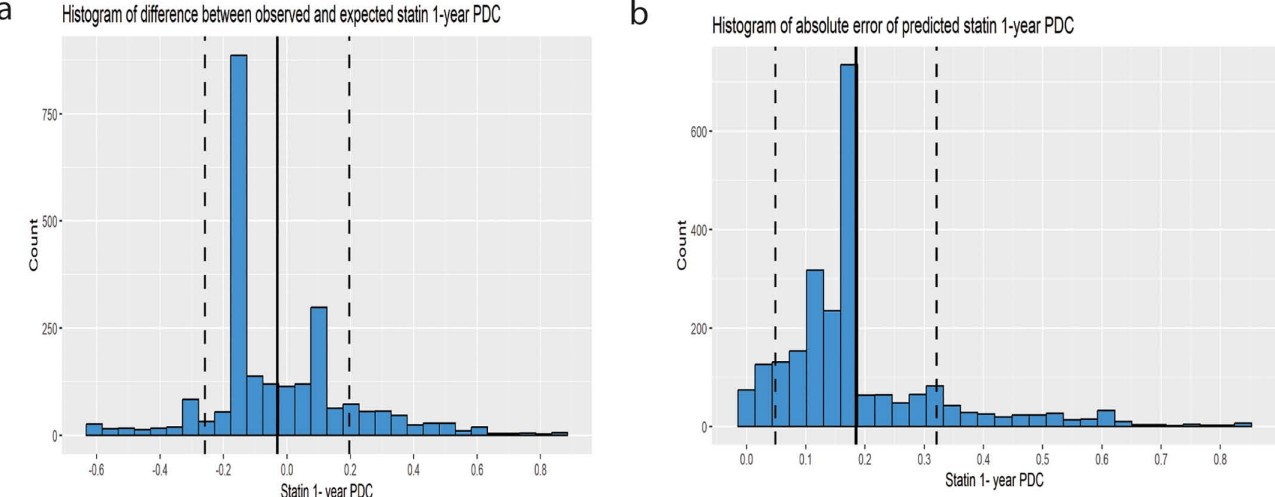

**Fig 2. Histogram of error distribution from the prediction model for statin.** Histogram of error distribution from the prediction model for statin 1-year PDC. **(a)** Histogram of the difference between the observed and expected statin 1-year PDC. The solid black vertical line represents the mean difference, and the dashed vertical lines represent one standard deviation from the mean. **(b)** Histogram of absolute error of predicted statin 1-year PDC. The solid black vertical line represents the mean difference, and the dashed vertical lines represent one standard deviation from the mean.

**Table 2. External validation results.**

| Model | R² | CITL | Calibration slope (95% CI) | RMSE | MAE |
|---|---|---|---|---|---|
| Statin | 0.67 | −0.037 | 1.06 (1.03,1.09) | 0.229 | 0.184 |
| Antiplatelet | 0.56 | −0.025 | 0.96 (0.93,1) | 0.255 | 0.202 |

$R^2$: proportion of explained variance, CITL; calibration-in-the-large, RMSE: Root Mean Squared Error, MAE: Mean Absolute Error.

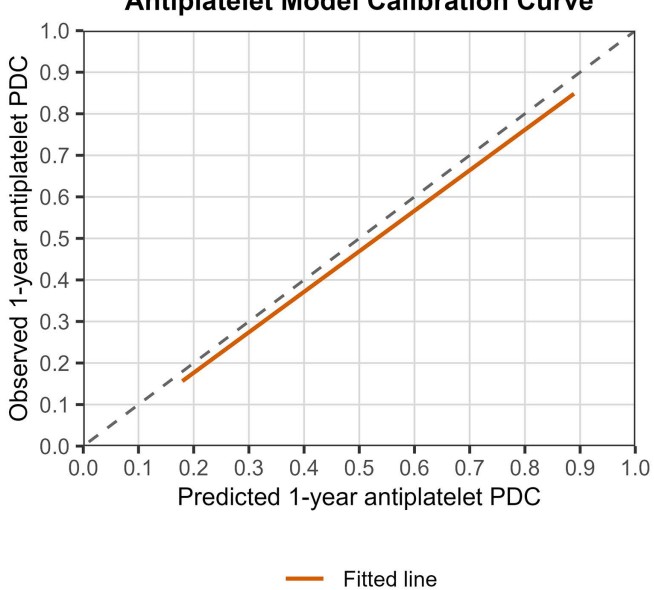

**Fig 3. Antiplatelet model calibration curve.** Calibration performance on external validation of the prediction model for antiplatelets 1-year PDC. The 45° dashed-line shows perfect calibration, and the blue line shows the predicted average PDC antiplatelets 1-year. The plot shows the observed outcome (y-axis) versus the predicted outcomes (x-axis).

## Discussion

Our current study involved external validation of two previously published models for statin and antiplatelet agents (aspirin and clopidogrel) in a cohort of patients who had experienced a stroke or TIA. The statin model showed an explained variance ($R^2$) of 0.67, meaning it accounted for two-thirds of the variance in the external dataset. The antiplatelet model had an explained variance ($R^2$) of 0.56, indicating it explained 56% of the variance in the external dataset. Additionally, the difference between the average observed 1-year PDC and the average expected (predicted) 1-year PDC was very small for both models (−3.7% for statin and −2.5% for antiplatelet). The calibration slope for both models was also close to 1, indicating a well-calibrated model. Overall, these results suggest that both models performed well in a new dataset of post-stroke patients.

External validation of adherence models is rarely conducted; in an extensive literature search, we found no external validation studies for medication adherence prediction models.

Median 1-year PDC was relatively low in our cohort for both statins and antiplatelets (41% and 59%, respectively). In comparison, Dalli et al. reported a median PDC of 85% for both medication groups in a study based on an Australian registry [7].

Although several studies have investigated predictors of medication adherence in post-stroke patients [25–28], few medication adherence models predicting adherence have been published for this patient group. Smoking status, number

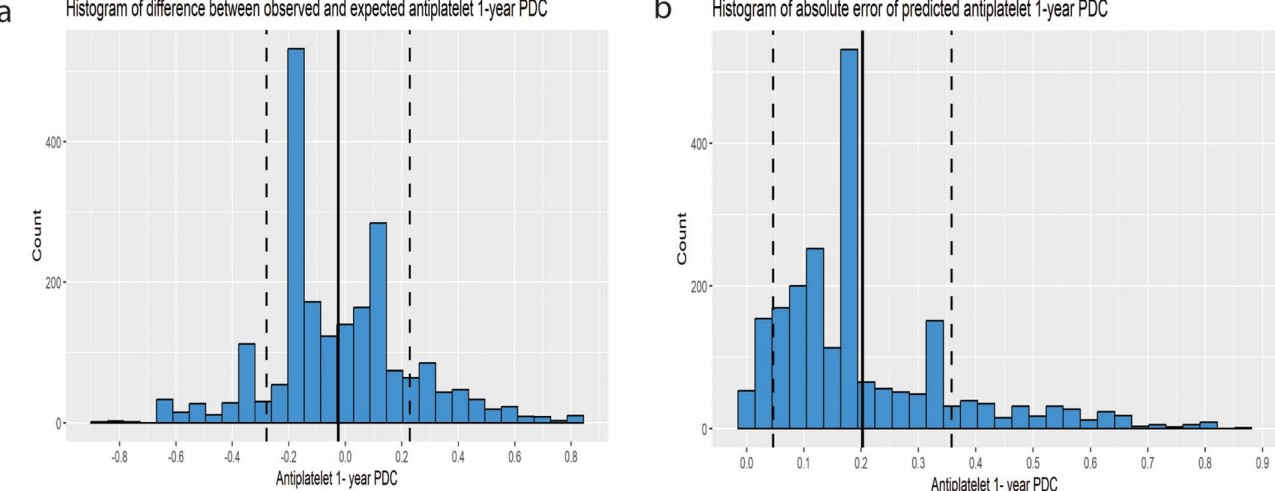

**Fig 4. Histogram of error distribution from the prediction model for antiplatelets.** Histogram of error distribution from the prediction model for antiplatelets 1-year PDC. **(a)** Histogram of the difference between observed and expected antiplatelets 1-year PDC. The solid black vertical line represents the mean difference, and the dashed vertical lines represent one standard deviation from the mean. **(b)** Histogram of absolute error of predicted antiplatelets 1-year PDC. The solid black vertical line represents the mean difference, and the dashed vertical lines represent one standard deviation from the mean.

of prescribed medicines at discharge, frequency of daily doses, beliefs about medication, and doctor-patient communication were found in some studies to be related to improved adherence [25–28]. However, in a previous study, we showed that these and other baseline predictors do not improve medication adherence prediction when the first 90 days' adherence data are included [14]. Furthermore, explained variability is very low when baseline factors alone are used for prediction, as compared to 90-day adherence data.

## Strength and limitations

The primary strength of this study lies in its design as an external validation of previously published models. The aim of the adherence prediction model is to be used in clinical practice, and external validation increases confidence in a prediction model. Nevertheless, external validation is not a single-step process [29], and the adherence prediction model may require adjustments and refinements that strike a balance between local performance and generalizability before being implemented in a specific setting. Therefore, validation in additional patient groups and ideally by another research group would indeed be needed to strengthen generalizability.

This study has several limitations. The major limitation of this study is its dependence on 1-month refill pattern. More than 94% of the patients included in this study had a 1-month supply of medication. Thus, the existing models may not be applicable in contexts where dispensing a 2- or 3-month supply is standard practice. This fact limits the international generalizability of the current models, especially in countries where 90-day fills are standard. To adapt the current models to settings where 2–3 months supplies are prevalent, the first 180 days adherence data need to be used as predictor instead of the first 90-days adherence data. Furthermore, LHCS is the smallest health provider in Israel, which may potentially limit model generalizability. However, LHCS members come from varied ethnic, geographic, and socioeconomic backgrounds. Consequently, the LHCS health provider's membership broadly represents the Israeli population. Another potential limitation is the assumption that prescription-fill data accurately represent patients' actual medication intake, which cannot be directly verified. Specifically, since low-dose aspirin is available over the counter in Israel, the

PDC for antiplatelet agents may have been underestimated. An additional potential limitation is that the MAE was relatively high (18.4% and 20.2% for statins and antiplatelets, respectively), due to the effect of relatively rare but extreme prediction errors on MAE. Nevertheless, the relative MAE reduction was high for both models (51.3% and 42.1% for statins and antiplatelets, respectively) suggesting that the model is useful for individual patient adherence prediction. Finally, in populations where the mean 1-year PDC of antiplatelets and statins differs significantly from our population, the model's performance may be inadequate. However, the model can be easily adapted to account for this difference before being used for adherence prediction by adding the difference in means between the two populations to the predicted values.

### Implications for practice

Our findings have two major implications. For clinicians, the present study demonstrates that 1-year adherence can be reliably predicted from data on the first 90 days of adherence. Moreover, our results confirm the value of therapeutic education in the first days and weeks post-stroke/TIA.

For health-care insurance providers, we recommend implementing the simplified model in patient electronic records to identify patients with low predicted PDC. For example, automated alerts that include the 1-year predicted PDC can be generated for the patient case manager. Additionally, patients with a low predicted 1-year PDC may be automatically connected to evidence-based mobile health interventions, such as phone calls, SMS text messaging, and telehealth [8]. Furthermore, we recommend habit formation enhancing interventions immediately after the event, and continuation of these interventions for at least 3 months.

## Conclusion

We performed an external validation of a previously published model for the prediction of 1-year PDC of statins and antiplatelets. The model performed well on a new patient population comprised of post-stroke patients. Moreover, our results provide further evidence that simple models based on the first 90 days' adherence data provide well-calibrated predictions and high agreement between predicted and observed outcome values on average.

## Supporting information

**S1 Table. TEN-SPIDERS reporting tool for PDC.**
(PDF)

**S2 Table. Summary of model comparisons.**
(PDF)

**S3 Table. Abbreviations.**
(PDF)

## Author contributions

**Conceptualization:** elias edward tannous, Shlomo Vinker, David Stepensky, Eyal Schwarzberg.

**Data curation:** elias edward tannous.

**Methodology:** elias edward tannous, Shlomo Vinker, David Stepensky, Eyal Schwarzberg.

**Supervision:** David Stepensky, Eyal Schwarzberg.

**Writing – original draft:** elias edward tannous.

**Writing – review & editing:** David Stepensky, Eyal Schwarzberg.

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
