## [Decision Letter · Decision Letter 0]

7 Oct 2025

PONE-D-25-37363Prediction of adherence to treatment with statins and anti-platelet drugs in first-year post-stroke patients: validation of beta-regression modelsPLOS ONE

Dear Dr. tannous,

Thank you for submitting your manuscript to PLOS ONE. After careful consideration, we feel that it has merit but does not fully meet PLOS ONE’s publication criteria as it currently stands. Therefore, we invite you to submit a revised version of the manuscript that addresses the points raised during the review process.

**Interesting study**

**Please address the comments if you are interested to resubmit**==============================

We look forward to receiving your revised manuscript.

Kind regards,

Yee Gary Ang, MBBS MPH

Academic Editor

PLOS ONE

Journal Requirements:

https://academic.oup.com/eurjpc/article-abstract/32/8/649/7811226?redirectedFrom=fulltext&login=false

In your revision ensure you cite all your sources (including your own works), and quote or rephrase any duplicated text outside the methods section. Further consideration is dependent on these concerns being addressed.

4. In the online submission form, you indicated that [The data underlying the results presented in the study are available from Leumit Health Care Services. https://www.leumit.co.il/contact-us/#].

Reviewers' comments:

Reviewer's Responses to Questions

**Comments to the Author**

1. Is the manuscript technically sound, and do the data support the conclusions?

Reviewer #1: Yes

2. Has the statistical analysis been performed appropriately and rigorously?

Reviewer #1: Yes

3. Have the authors made all data underlying the findings in their manuscript fully available?

Reviewer #1: Yes

4. Is the manuscript presented in an intelligible fashion and written in standard English?

Reviewer #1: Yes

5. Review Comments to the Author

Reviewer #1: Attachment Major Comments

1. Central premise of the model and prescription cycle dependence

o The main novelty of this work is that adherence in the first 90 days serves as the sole predictor of 1-year adherence, eliminating the need for additional demographic or socioeconomic variables. This is an elegant and clinically appealing concept, and it likely reflects habit formation.

o However, this premise only holds in systems with 30-day prescription cycles, where patients have three refill opportunities in the first 90 days, allowing meaningful assessment of adherence patterns.

o In systems where 90-day fills are standard (e.g., North America, many European countries), the “90-day adherence” measure collapses to a near-binary indicator (one fill vs. no fill). This strips away predictive resolution, meaning the model could become substantially less useful, or even unusable, in those contexts.

o I strongly recommend that the authors:

Explicitly acknowledge that the model is tied to 30-day fill patterns.

Discuss in both the Introduction and Discussion how this limitation affects international generalizability.

Suggest how the model might need to be recalibrated or adapted (e.g., shorter observation windows, refill consistency measures) for settings with 90-day fills.

6. PLOS authors have the option to publish the peer review history of their article (what does this mean?). If published, this will include your full peer review and any attached files.

Reviewer #1: No

---

## [Author Response · Author response to Decision Letter 1]

4 Nov 2025

we have ensured style requirements.

2. Please note that PLOS One has specific guidelines on code sharing for submissions in which author-generated code underpins the findings in the manuscript. In these cases, we expect all author-generated code to be made available without restrictions upon publication of the work.

Code is available without restriction.

https://academic.oup.com/eurjpc/article-abstract/32/8/649/7811226?redirectedFrom=fulltext&login=false

The current study is an external validation of the models developed in this publication: https://academic.oup.com/eurjpc/article-abstract/32/8/649/7811226?redirectedFrom=fulltext&login=false. We have cited this work many times in the manuscript.

4. In the online submission form, you indicated that [The data underlying the results presented in the study are available from Leumit Health Care Services. https://www.leumit.co.il/contact-us/#].

We have revised this statement. The data underlying the results presented in the study are available by request to the corresponding author and after permission by Leumit Health Care Services. Making the data publicly available would breach compliance with the protocol approved by the research ethics board. However, for the main results (model calibration and model performance) the data can be shared by contact with the corresponding author and after permission by LHCS. tanus@bgu.ac.il

Reviewers comments: we have addressed all reviewr comments in the "Response to Reviewers" file.

---

## [Decision Letter · Decision Letter 1]

5 Jan 2026

PONE-D-25-37363R1Prediction of adherence to treatment with statins and anti-platelet drugs in first-year post-stroke patients: validation of beta-regression modelsPLOS One

Dear Dr. tannous,

Thank you for submitting your manuscript to PLOS ONE. After careful consideration, we feel that it has merit but does not fully meet PLOS ONE’s publication criteria as it currently stands. Therefore, we invite you to submit a revised version of the manuscript that addresses the points raised during the review process.

**ACADEMIC EDITOR:**

Happy New Year

One of the reviewer has recommended further improvements

Please see and address if possible.

We look forward to receiving your revised manuscript.

Kind regards,

Yee Gary Ang, MBBS MPH

Academic Editor

PLOS One

Journal Requirements:

Reviewers' comments:

Reviewer's Responses to Questions

**Comments to the Author**

1. If the authors have adequately addressed your comments raised in a previous round of review and you feel that this manuscript is now acceptable for publication, you may indicate that here to bypass the “Comments to the Author” section, enter your conflict of interest statement in the “Confidential to Editor” section, and submit your "Accept" recommendation.

Reviewer #1: All comments have been addressed

Reviewer #2: All comments have been addressed

2. Is the manuscript technically sound, and do the data support the conclusions?

Reviewer #1: Yes

Reviewer #2: Partly

3. Has the statistical analysis been performed appropriately and rigorously?

Reviewer #1: Yes

Reviewer #2: Yes

4. Have the authors made all data underlying the findings in their manuscript fully available?

Reviewer #1: No

Reviewer #2: Yes

5. Is the manuscript presented in an intelligible fashion and written in standard English?

Reviewer #1: Yes

Reviewer #2: Yes

6. Review Comments to the Author

Reviewer #1: Reviewer Response

I thank the authors for their detailed and thoughtful responses to my comments, as well as for the extensive revisions made to the manuscript. I have carefully reviewed the updated text, figures, and the point-by-point rebuttal.

Overall, the authors have addressed my major and minor comments thoroughly and satisfactorily.

1. Prescription-cycle dependence and international generalizability

The authors have now clearly acknowledged the dependence on 30-day fill patterns and have added appropriate explanation in both the Introduction and Discussion. The proposed extension to a 180-day observation window for systems with 90-day fills is sensible and enhances clarity regarding transportability. This substantially strengthens the manuscript.

2. Predictive performance interpretation

The expanded discussion of explained variance (R²), clinical usefulness, and the value of incorporating decision-analytic perspectives (e.g., decision curve analysis) improves transparency and interpretability. The clarification around the relative MAE reduction is particularly helpful.

3. Introduction and literature context

The Introduction has been updated.

4. Stroke vs. TIA populations

The rationale for analyzing stroke and TIA patients together is now justified.

5. Generalizability and LHCS dataset

The revised Methods now clearly indicate the coverage of LHCS (7.6% of the Israeli population) and justify representativeness. The authors appropriately highlight that further validation in independent datasets and other countries is warranted.

6. Calibration and error metrics

The authors appropriately contextualized the magnitude of prediction error and clarified the difference between absolute error and relative MAE reduction. The added discussion improves the reader’s ability to judge potential applications at the individual vs. population level.

7. PDC limitations

The manuscript now clearly discusses the inherent limitations of PDC, including aspirin availability over the counter and the fact that dispensation does not guarantee ingestion.

8. Implications for practice

The expanded section on implementation within electronic health records—including automated alerts and linkage to mHealth adherence interventions—is clear, practical, and referenced.

9. Minor issues

Typographical errors, table formatting, terminology clarification, and figure readability have been addressed.

Overall Recommendation

The manuscript has been substantially improved.

Reviewer #2: Abstract concludes that the model “may be used for adherence prediction,” which is accurate but vague. Given that this is an external validation study, readers expect at least a brief indication of how the model would be operationalized in practice (e.g., early identification of high-risk patients within 90 days post-stroke).

Introduction relies heavily on the authors’ prior myocardial infarction study as the conceptual backbone of the current work. While this is understandable given the study design, it subtly shifts the narrative from hypothesis-driven inquiry to confirmatory extension.

There are subtle methodological choices that deserve more explicit discussion. The overwhelming predominance of 1-month refill patterns simplifies modeling but also constrains generalizability. While this is acknowledged later as a limitation, the methods section could benefit from explicitly stating that the model’s structure is intrinsically linked to refill cadence. The absence of identified bleeding events in the antiplatelet cohort is another point that requires clarification.

Results section largely reports metrics without interrogating their implications. High R² values are expected when early adherence is used to predict later adherence, yet this inherent predictability is not explicitly acknowledged.

7. PLOS authors have the option to publish the peer review history of their article (what does this mean?). If published, this will include your full peer review and any attached files.

Reviewer #1: No

Reviewer #2: No

---

## [Author Response · Author response to Decision Letter 2]

7 Jan 2026

We thank the reviewers for their insightful and helpful comments.

Response for Reviewer #1:

We thank reviewer #1 for the thorough and thoughtful review.

Response for Reviewer #2:

Abstract concludes that the model “may be used for adherence prediction,” which is accurate but vague. Given that this is an external validation study, readers expect at least a brief indication of how the model would be operationalized in practice (e.g., early identification of high-risk patients within 90 days post-stroke).

We appreciate the reviewer for this valuable recommendation. The abstract has been revised to clearly state that the model can be used for early detection of patients at increased risk of non-adherence within the first 90 days after a stroke, helping to enable prompt and targeted adherence-support interventions.

Introduction relies heavily on the authors’ prior myocardial infarction study as the conceptual backbone of the current work. While this is understandable given the study design, it subtly shifts the narrative from hypothesis-driven inquiry to confirmatory extension.

We appreciate this observation. As the reviewer pointed out, since the study serves as an external validation of previously developed models, it relies heavily on our earlier published work. However, this is not the only reason. We conducted an extensive literature review and found very few studies focusing on predictive modeling of medication adherence in post-TIA/stroke patients. Additionally, no other studies utilizing beta-regression for adherence prediction were identified. These are the primary reasons we heavily relied on our previous study in the introduction.

There are subtle methodological choices that deserve more explicit discussion. The overwhelming predominance of 1-month refill patterns simplifies modeling but also constrains generalizability. While this is acknowledged later as a limitation, the methods section could benefit from explicitly stating that the model’s structure is intrinsically linked to refill cadence. The absence of identified bleeding events in the antiplatelet cohort is another point that requires clarification.

We agree that these methodological features need a clearer description in the Methods section. We revised the methods to explicitly state that the model structure is directly connected to the monthly refill schedule, reflecting the dominant pattern of 1-month dispensing in the dataset .

Regarding bleeding events, we clarified in the results section that no bleeding events were identified based on available ICD-9 documentation. This may reflect limited sensitivity rather than a true absence. We also acknowledged that minor bleeding events could have occurred but were not serious enough for patients to seek medical advice.

Results section largely reports metrics without interrogating their implications. High R² values are expected when early adherence is used to predict later adherence, yet this inherent predictability is not explicitly acknowledged.

We thank the reviewer for this comment. We expanded on the implications of the metrics in the results section. We appreciate the reviewer’s comment and agree that high R² values are expected when early adherence is used to predict later adherence. We added an explicit acknowledgment of this fact to the results section. We also cited previous studies showing that adding early adherence to prediction models improves model performance. However, in the present study, which is an external validation study, a high R² value is not trivial since we did not fit a new model to the data but tested a previously developed model. Thus, a high R² in the external validation suggests that the mean relation between early adherence and 1-year adherence is relatively stable across patient populations. Typically, even well-performing models with high R² in the development dataset show significant reductions in R² when validated externally. Therefore, the high R² in the external validation dataset indicates a stable predictor-outcome relationship across populations and demonstrates that the model is transportable. We added a summary of this discussion to the results section.

---

## [Decision Letter · Decision Letter 2]

9 Feb 2026

PONE-D-25-37363R2Prediction of adherence to treatment with statins and anti-platelet drugs in first-year post-stroke patients: validation of beta-regression modelsPLOS One

Dear Dr. tannous,

Thank you for submitting your manuscript to PLOS ONE. After careful consideration, we feel that it has merit but does not fully meet PLOS ONE’s publication criteria as it currently stands. Therefore, we invite you to submit a revised version of the manuscript that addresses the points raised during the review process.

**ACADEMIC EDITOR:**

We have invited several reviewers

2 have reverted but they have some further comments to be addressed.

Please see and resubmit if you are able to address them.

We look forward to receiving your revised manuscript.

Kind regards,

Yee Gary Ang, MBBS MPH

Academic Editor

PLOS One

Journal Requirements:

Reviewers' comments:

Reviewer's Responses to Questions

**Comments to the Author**

1. If the authors have adequately addressed your comments raised in a previous round of review and you feel that this manuscript is now acceptable for publication, you may indicate that here to bypass the “Comments to the Author” section, enter your conflict of interest statement in the “Confidential to Editor” section, and submit your "Accept" recommendation.

Reviewer #3: All comments have been addressed

Reviewer #4: (No Response)

2. Is the manuscript technically sound, and do the data support the conclusions?

Reviewer #3: Yes

Reviewer #4: Partly

3. Has the statistical analysis been performed appropriately and rigorously?

Reviewer #3: Yes

Reviewer #4: I Don't Know

4. Have the authors made all data underlying the findings in their manuscript fully available?

Reviewer #3: Yes

Reviewer #4: No

5. Is the manuscript presented in an intelligible fashion and written in standard English?

Reviewer #3: Yes

Reviewer #4: Yes

6. Review Comments to the Author

Reviewer #3: (No Response)

Reviewer #4: Interesting manuscript that may be useful as clinicians and health insurers try to predict adherence in these population groups. Below are a few suggestions for improving the manuscript:

1. Line 288: these patients with 2 and 3-month days supply were still included in the validation then?

2. Line 299: can you put the socioeconomic status in context, ie. were most patients middle-class then?

3. Line 301: can you provide any explanation for why the PDC is so low? In the limitations you note that antiplatelets may be underestimated due to over-the-counter meds, but this is the statin cohort. Also why does PDC get better over time, wouldn't you expect it to be highest in the first 90 days and then trail off? Also by "new users" you mean these patients were not on statins prior to experiencing stroke? Related, why do you think PDC was so much higher for the antiplatelet cohort?

4. Line 435: In what country, can you put these results in context?

5. Line 443: typo should be medication not mediation

6. Line 487: this isn't really supported by the results of the current study looking at external validation, rather the baseline variables being poor predictors of adherence was reported in a prior study.

7. Figures are somewhat unclear and should be higher quality

7. PLOS authors have the option to publish the peer review history of their article (what does this mean?). If published, this will include your full peer review and any attached files.

Reviewer #3: **Yes:**Giovanni Merlino

Reviewer #4: No

---

## [Author Response · Author response to Decision Letter 3]

6 Mar 2026

We thank the reviewer for these valuable comments and suggestions.

1. Line 288: these patients with 2 and 3-month days supply were still included in the validation then?

Yes, these patients were included. Since there were relatively very few patients who had a 2 or 3-month supply, we highlighted the fact that these models are more suitable for a 1-month supply. But in the validation cohort we included the (few) patients with 2 and 3-month.

2. Line 299: can you put the socioeconomic status in context, ie. were most patients middle-class then?

We thank the reviewer for this comment. Yes, most patients could be considered middle class. We added this to the results section.

3. Line 301: can you provide any explanation for why the PDC is so low? In the limitations you note that antiplatelets may be underestimated due to over-the-counter meds, but this is the statin cohort. Also why does PDC get better over time, wouldn't you expect it to be highest in the first 90 days and then trail off? Also by "new users" you mean these patients were not on statins prior to experiencing stroke? Related, why do you think PDC was so much higher for the antiplatelet cohort?

We thank the reviewer for these comments.

can you provide any explanation for why the PDC is so low?

Unfortunately, we didn`t find studies that dealt with the question of the underlying reasons of non-adherence or lower adherence to statins post TIA/stroke. In contrast, in post-myocardial infarction patients, the median was higher (PDC of 0.8). (1) We can only speculate that this may be related to how patients perceive the risk of recurrence in these two different indications.

1-Tannous EE, Selitzky S, Vinker S, Stepensky D, Schwarzberg E. Predictive modelling of medication adherence in post-myocardial infarction patients: a Bayesian approach using beta-regression. Eur J Prev Cardiol. 2025;32(8):649-658. doi:10.1093/eurjpc/zwae327

Also why does PDC get better over time, wouldn't you expect it to be highest in the first 90 days and then trail off?

The difference is between the median PDC at 90 days and the median PDC at 1 year, primarily in the antiplatelet cohort. The main reason is that in the antiplatelet group, some patients started filling aspirin several months after the event. While the expected pattern is high PDC initially then tapering off, the adherence literature shows a group of patients characterized by early decline and partial recovery. (2)

2-Salmasi S, De Vera MA, Safari A, et al. Longitudinal Oral Anticoagulant Adherence Trajectories in Patients With Atrial Fibrillation. J Am Coll Cardiol. 2021;78(24):2395-2404. doi:10.1016/j.jacc.2021.09.1370

? Also by "new users" you mean these patients were not on statins prior to experiencing stroke?

Yes, “new users” are patients who were not on statins prior to experiencing stroke as opposed to “prevalent users”.

Related, why do you think PDC was so much higher for the antiplatelet cohort?

Here, we can only speculate about the reason. We think one possibility is that patients perceive the underlying cause of the stroke/TIA as related to blood clots, and therefore believe that antiplatelets are more important than statins in this context.

4. Line 435: In what country, can you put these results in context?

We appreciate the reviewer's comment. The referenced study used data from Australia. We have added this clarification to the manuscript.

5. Line 443: typo should be medication not mediation

We thank the reviewer for this comment. We corrected the typo.

6. Line 487: this isn't really supported by the results of the current study looking at external validation, rather the baseline variables being poor predictors of adherence was reported in a prior study.

We thank the reviewer for this comment. We agree and have revised the sentence to “the present study demonstrates that 1-year adherence can be reliably predicted from data on the first 90 days of adherence”.

7. Figures are somewhat unclear and should be higher quality

We thank the reviewer for this comment. We increased the dpi from 300 to 600.

---

## [Decision Letter · Decision Letter 3]

13 Mar 2026

Prediction of adherence to treatment with statins and anti-platelet drugs in first-year post-stroke patients: validation of beta-regression models

PONE-D-25-37363R3

Dear Dr. tannous,

We’re pleased to inform you that your manuscript has been judged scientifically suitable for publication and will be formally accepted for publication once it meets all outstanding technical requirements.

Kind regards,

Yee Gary Ang, MBBS MPH

Academic Editor

PLOS One

Additional Editor Comments (optional):

Reviewers' comments:

Reviewer's Responses to Questions

**Comments to the Author**

1. If the authors have adequately addressed your comments raised in a previous round of review and you feel that this manuscript is now acceptable for publication, you may indicate that here to bypass the “Comments to the Author” section, enter your conflict of interest statement in the “Confidential to Editor” section, and submit your "Accept" recommendation.

Reviewer #4: All comments have been addressed

2. Is the manuscript technically sound, and do the data support the conclusions?

Reviewer #4: Yes

3. Has the statistical analysis been performed appropriately and rigorously?

Reviewer #4: I Don't Know

4. Have the authors made all data underlying the findings in their manuscript fully available?

Reviewer #4: Yes

5. Is the manuscript presented in an intelligible fashion and written in standard English?

Reviewer #4: Yes

6. Review Comments to the Author

Reviewer #4: (No Response)

7. PLOS authors have the option to publish the peer review history of their article (what does this mean?). If published, this will include your full peer review and any attached files.

Reviewer #4: No

---

## [Editor Report · Acceptance letter]

PONE-D-25-37363R3

PLOS One

Dear Dr. tannous,

I'm pleased to inform you that your manuscript has been deemed suitable for publication in PLOS One. Congratulations! Your manuscript is now being handed over to our production team.

Kind regards,

on behalf of

Dr. Yee Gary Ang

Academic Editor

PLOS One